# Model guided trait-specific co-expression network estimation as a new perspective for identifying molecular interactions and pathways

**Juho A. J. Kontio**[1], **Tanja Pyhäjärvi**[2,3], **Mikko J. Sillanpää**[1] *

**1** Research Unit of Mathematical Sciences, University of Oulu, Oulu, Finland, **2** Department of Ecology and Genetics, University of Oulu, Oulu, Finland, **3** Department of Forest Sciences, University of Helsinki, Helsinki, Finland

* mikko.sillanpaa@oulu.fi

**Data Availability Statement:** The authors state that all data necessary for confirming the conclusions are represented fully within the article. Software and simulated replicates are presented in

## Abstract

A wide variety of 1) parametric regression models and 2) co-expression networks have been developed for finding gene-by-gene interactions underlying complex traits from expression data. While both methodological schemes have their own well-known benefits, little is known about their synergistic potential. Our study introduces their methodological fusion that cross-exploits the strengths of individual approaches via a built-in information-sharing mechanism. This fusion is theoretically based on certain trait-conditioned dependency patterns between two genes depending on their role in the underlying parametric model. Resulting trait-specific co-expression network estimation method 1) serves to enhance the interpretation of biological networks in a parametric sense, and 2) exploits the underlying parametric model itself in the estimation process. To also account for the substantial amount of intrinsic noise and collinearities, often entailed by expression data, a tailored co-expression measure is introduced along with this framework to alleviate related computational problems. A remarkable advance over the reference methods in simulated scenarios substantiate the method's high-efficiency. As proof-of-concept, this synergistic approach is successfully applied in survival analysis, with acute myeloid leukemia data, further highlighting the framework's versatility and broad practical relevance.

## Author summary

Here we built up a mathematically justified bridge between 1) parametric approaches and 2) co-expression networks in light of identifying molecular interactions underlying complex traits. We first shared our concern that methodological improvements around these schemes, adjusting only their power and scalability, are bounded by more fundamental scheme-specific limitations. Subsequently, our theoretical results were exploited to overcome these limitations to find gene-by-gene interactions neither of which can capture alone. We also aimed to illustrate how this framework enables the interpretation of co-

the S2 Appendix and GitHub repository https://github.com/JAJKontio/model_diffnet - released under the \GNU General Public License v3.0. The analyzed DREAM9 acute myeloid leukemia dataset is available upon registration at https://www.synapse.org/#!Synapse:syn2455683/wiki/64007. The validation cohort was downloaded from the TCGA (The Cancer Genome Atlas) database (LAML - https://cancergenome.nih.gov/) and used also via the online GEPIA-software at http://gepia.cancer-pku.cn/.

**Funding:** JK was supported by the Finnish Academy of Science and Letters, Vilho, Yrjö and Kalle Väisälä Foundation grant nr. 190030 URL https://www.acadsci.fi/apurahat-ja-palkinnot/haettavat-apurahat/vaisalan-rahasto.html and Biocenter Oulu URL https://www.oulu.fi/biocenter/. TP acknowledges Academy of Finland grant nr. 287431 URL https://www.aka.fi/en/. MJS acknowledges Academy of Finland (PROFI5 HiDyn) grant nr. 326291 URL https://www.aka.fi/en/ and is supported by the Infotech Oulu research institute (https://www.oulu.fi/infotech/). The funders had no role in study design, data collection and analysis, decision to publish, or preparation of the manuscript.

**Competing interests:** The authors have declared that no competing interests exist.

expression networks in a more parametric sense to achieve systematic insights into complex biological processes more reliably. The main procedure was fit for various types of biological applications and high-dimensional data to cover the area of systems biology as broadly as possible. In particular, we chose to illustrate the method's applicability for gene-profile based risk-stratification in cancer research using public acute myeloid leukemia datasets.

This is a *PLOS Computational Biology* Methods paper.

## Introduction

Gene-by-gene interactions are known to underlie phenotypes in a variety of systems [1, 2]. A huge amount of research has been devoted to robust identification of such components from high-throughput biological data [3–5]. The development of this methodology is mainly focused on exhaustive search approaches and implementing algorithms that alleviate their computational complexities [4, 6, 7]. Typically, these methods are restricted into parametric models consisting of overly simplified interactions types (e.g. product terms) contributing additively to the phenotype [8]. However, gene-by-gene interactions are interpreted biologically more broadly than parametric models often allow, such as functional interactions between genes in biological pathways [1, 2, 8, 9].

To identify functionally related genes or members of the same pathway from omics data, one could benefit from the vast scheme of co-expression network analysis [10]. In co-expression networks, each node represents a single gene, and is connected with another nodes if the expression values of the corresponding genes are dependent. In particular, there has been a growing interest on estimating simultaneously two co-expression networks, such that the estimation process accounts for some external state of interest [11–17]. For instance, in transcriptional interactions, where a transcription factor binds to promoter regions of a particular gene to regulate its expression levels, can be disrupted in cancers [12, 18]. Then co-expression networks estimated separately over case and control samples are expected to have interesting differences due to the cancer-specific dysregulations in transcriptional mechanisms.

Co-expression networks can be estimated from data (in both case-control and single populations) either with unconditional or conditional dependency metrics. A popular unconditional approach is to measure, in all its simplicity, just a pairwise correlation/covariance between genes [19–21]. However, this simple metric has received a lot of criticism for evaluating direct dependencies—false positive edges between genes might occur in the presence of confounding factors [22, 23]. For this problem, inverse-covariance matrix based Gaussian graphical models (GGMs) [24, 25] provide a complete solution, as they are capable of distinguishing direct relationships from indirect ones [26, 27].

These network comparison procedures are often motivated by the deficiencies of the exhaustive search approaches. On the other hand, the current network approaches are lacking some very crucial properties of the parametric interaction models in turn. These include, for instance, an explicit connection with the trait of interest, intuitive parametric interpretations, various options for hypothesis testing, and a possibility to account for the main effects. Despite the popularity of differential co-expression network analyses, these critical issues have remained unresolved.

Little is also known about their synergistic potential in the search of molecular interactions and pathways underlying complex traits. That is, a mathematical presentation, that would formalize the relationship between co-expression networks and parametric interaction models remains undefined. Here we consider the issues above and characterize the relationship between these methodological schemes. Finally, a methodological fusion is provided to cross-exploit all scheme-specific strengths via a built-in information-sharing mechanism. As proof-of-concept, the framework is applied for searching prognostically important gene-by-gene interactions in acute myeloid leukemia (AML).

## Results

Our approach combines the benefits of two popular methodological schemes, co-expression networks [10], and parametric gene-by-gene interaction models [3, 4], to find molecular interactions and pathways regulating complex traits neither of which can capture alone (Fig 1). This is based on three simple steps:

- Step 1: A pre-defined underlying parametric regression model for trait variation is first used to estimate such effects of genes that are not identifiable with co-expression networks (e.g. the main effects). The remaining unexplainable variation is then subtracted from the original trait variable and used as a new response variable for the next step.

- Step 2: The provided trait-specific co-expression network estimation method is then applied to estimate network structures that can explain the trait variation remained from the first step. A novel dependency metric is also provided to account for certain collinearities in data that are generally considered problematic with the parametric methods used in the first step.

- Step 3: The underlying parametric model is then used again to provide a parametric interpretation for the estimated co-expression network elements, which is not possible standalone from the co-expression networks.

While a high number of co-expression techniques have been introduced over the last 20 years, and parametric regression model type approaches even longer, the synergistic potential of these different schemes has remained unnoticed. The above three steps fuses these two schemes into a one, easy-to-use model guided co-expression network estimation procedure, to overcome the limitations of each. In particular, these model guided co-expression networks are always response variable specific by the construction. This means that all relationship between genes, whose interplay is not important for the response variable of interest, are excluded from the resulting network. In advance of illustrating its usage in the simulated and real examples, we briefly summarize these two schemes and outline the proposed fusion itself.

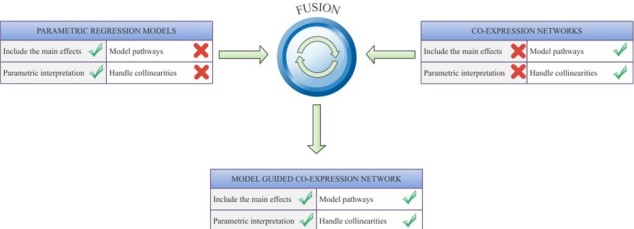

**Fig 1. Schematic overview.** A diagram of the conceptual entities of the proposed fusion (model guided co-expression networks) combining two schemes—parametric interaction models and co-expression networks. The most central methodological benefits (green checkmarks) and deficiencies (red X marks) are listed method-specifically.

## Gene-by-gene interaction model

In a regular gene-by-gene interaction model, normally distributed (with zero means) expression levels of individual genes $X_{i1}, \ldots, X_{ip}$ as well as the pairwise interactions between them are associated additively with the quantitative phenotype $Y_i$ for an individual $i$ as follows:

$$Y_i = \mu + \sum_{j=1}^{p} X_{ij}\beta_j + \sum_{k>j} X_{ij}X_{ik}\beta_{jk} + \varepsilon_i, \tag{1}$$

where $\varepsilon_i \stackrel{\text{i.i.d}}{\sim} \mathcal{N}(0, \sigma^2)$ for all $i = 1, \ldots, n$. Here $\mu$ is the population intercept and later assumed to be zero, $\beta_j$ is the main effect of $j$th gene $X_j$ and $\beta_{jk}$ is the effect of gene-by-gene interaction term between genes $X_j$ and $X_k$. A subindex $i$ in each gene represents the expression value of the corresponding gene measured from an individual $i$. In particular, we will refer to the interaction terms of the model (1) as type I interactions.

Despite its popularity, this model ignores completely the complex gene-gene interactions effects that do not contribute linearly to the phenotypic variation, e.g., through functional interactions between genes in biological pathways [1, 2, 8, 9]. Therefore we formulate an extension of the model (1) involving more complex activation/deactivation patterns between gene-expression levels that are associated only with either low or high phenotype values and less efficiently identifiable with parametric interaction model (1). To model such effects, we use an additional mapping $\Delta(\cdot, \cdot) : \mathbb{R}^2 \to \mathbb{R}$ which refers to an arbitrary gene activation/deactivation function yielding the following extension of the interaction model (1):

$$Y_i = \sum_{j=1}^{p} X_{ij}\beta_j + \sum_{k>j} X_{ij}X_{ik}\beta_{jk} + \sum_{k>j} \alpha_{jk}\Delta_{jk}(X_{ij}, X_{ik}) + \varepsilon_i. \tag{2}$$

For instance, in transcriptional interactions where a transcription factor binds to promoter regions of a particular gene to regulate its expression levels can be disrupted in cancers [12, 18]. Such gene-gene interactions $\Delta_{jk}(X_{ij}, X_{ik})$ will be referred as type II interactions.

## Co-expression networks

Generally, a probabilistic network $G$ refers to a pair $(V, E)$ of nodes $V := \{1, \ldots, p\}$ and the collection $E$ of edges connecting these individual nodes [24]. In co-expression networks, nodes denote a set of random variables $\{X_1, X_2, \ldots, X_p\}$ that correspond to the expression levels of $p$ individual genes. The collection $E$ of edges, in turn, represents a desired type of dependencies between individual genes. The aim of the co-expression network inference is to estimate these dependencies from data with a case-specifically chosen co-expression measure. For instance, simple covariance or correlation coefficients are the most popular co-expression measures for constructing co-expression networks as they are easy to estimate from data [10]. Alternatively, if a random vector, representing the expression levels of $p$ individual genes $\{X_1, X_2, \ldots, X_p\}$ (or its normalized version) follows a multivariate normal distribution $\mathcal{N}(0, \Sigma)$, co-expression networks are often modeled with the inverse covariance matrix $\Sigma^{-1}$ [24]. The latter is closely related to Gaussian graphical models (GGMs) which are capable of distinguishing direct relationships between genes from indirect ones [24], and are referred as such in the forthcoming sections.

## Problems in interaction search and new perspectives

Since popular exhaustive search methods [3–5] focus on the model (1), they are incapable of identifying type II interactions. These approaches are also often struggling with identifiability

issues caused by strongly co-expressed genes. Such deficiencies may cause highly incomplete and distorted conclusions about the role and proportion of gene-gene interactions in overall trait regulation mechanisms. Consequences might be particularly adverse in medical applications, e.g., when designing personalized treatments based on gene expression profiling. Nevertheless, both of these deficiencies are stemming from the fact that exhaustive search approaches are inefficient to account for the dependencies between genes.

In this regard, one could benefit from the vast co-expression network estimation methodology [19–21, 24, 25] designed exactly for such purposes. On the other hand, co-expression networks, in turn, are not well-suited for explicitly modeling various types of effects on phenotypic variation (e.g. the main or interaction effects) in comparison to the parametric regression models. As such, an appropriately implemented hybrid perspective is required for revealing interactions underlying complex traits efficiently. In the materials and methods section, we contribute to this area by fusing the parametric interaction model (2) into the co-expression network estimation in accordance to the following outline:

- Objective A: Incorporating the phenotypic information into the network estimation such that the network represents only phenotypically important gene relationships.

- Objective B: Determining an explicit link between the concept of co-expression networks and the generalized interaction model (2).

- Objective C: Exploiting the above link to derive a network estimation method in which type I and type II interactions are separated in the estimation process.

- Objective D: Introducing an estimation metric that accounts for the common characteristics of phenotype regulating mechanisms and expression data.

Now we have a framework for trait-specific co-expression network estimation based on parallel consideration and exploitation of the underlying interaction model (2). Particularly, this network provides evidence of gene-by-gene interactions (type I and II) in relation to the underlying parametric interaction model, while the co-expression networks generally represent only associations between genes. Thus, the estimated network connections are referred as interactions rather than associations in this context.

## A technical overview of the procedure

Here we give an overview of each step of the proposed method (see also Fig 2) which is presented and explained comprehensively in the materials and methods section.

**Step 1**. **Estimate the residual vector from the main effect model**
As a partial solution to the objective (D), the main effects are first estimated in the model (2) without the interaction terms to get the residual estimates

$$\hat{\varepsilon}_i = Y_i - \hat{\mu} - \sum_{j=1}^{p} X_{ij}\hat{\beta}_j. \tag{3}$$

Individuals are then divided into the high and low groups based on the empirical quantiles $Q\hat{\varepsilon}_i(a)$ and $Q\hat{\varepsilon}_i(1-a)$ $(a \in \,]0, 0.5])$ of the estimated residual values $\hat{\varepsilon}_i$ and two separate networks are estimated corresponding to these groups in the next step.

**Step 2**. **Estimate the high and low networks**
In this step, we first estimate two networks: A high network is constructed over individuals in a high group, i.e., to whom $\hat{\varepsilon}_i \geq Q\hat{\varepsilon}_i(1-a)$ and a low network is estimated over

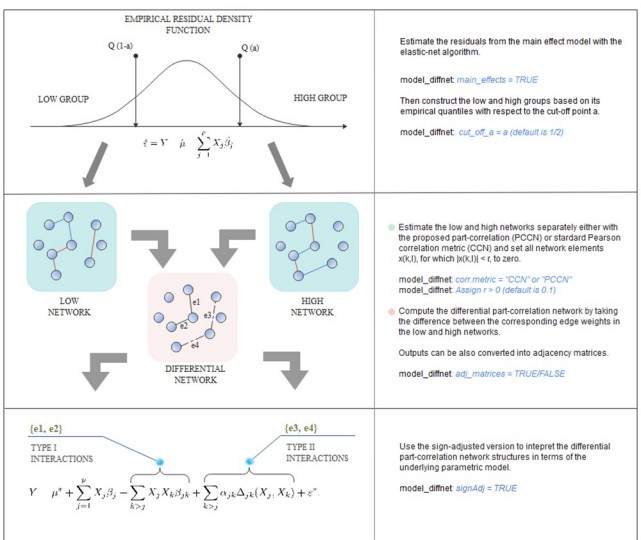

**Fig 2. Procedural workflow diagram.** A schematic representation of the logical structure underlying the provided algorithmic implementation. The first panel from the top represents the empirical density function of the residual vector estimated from the main effect model. Here $Q(\cdot)$ is a quantile function of this distribution and $a \in [0.5, 1[$ is a user-defined cut-off point. The second panel from the top illustrates how the differential networks are computed from the low and high networks using the sign-adjusted version. Different edge colors represent the edge weights of opposite signs, but equal magnitudes, for simplicity. The bottom panel demonstrates how different sign-combinations can be interpreted as type I or II interactions in the underlying parametric model. Moreover, the most important arguments of the provided model_diffnet R-function are explained in the right panel to indicate their role in this procedural flow.

individuals to whom $\hat{\varepsilon}_i \leq Q\hat{\varepsilon}_i(a)$ holds, respectively. To estimate the corresponding edge weights by accounting for the objective (D), we propose a novel correlation measure and compute the corresponding matrices $\Phi^a$ and $\Phi^{1-a}$ in the low and high groups. These matrices are then used to define a novel differential network structure $Q_a = |\Phi^{1-a} - \Phi^a|$, referred as differential part-correlation co-expression network (dPCCN), which links this framework to the objective (A).

**Step 3**. **Link the estimated networks structures to the underlying parametric model**

To further identify and separate type I and II interactions in the estimation process (objective C), we formulate its sign-adjusted version $Q_{\text{sgn,a}}$, that uses a set of rules to characterize each differential network element as negative, zero or positive. Each of them denotes a specific parametric interpretation, and builds up a connection between the differential network elements and the effects in the parametric interaction model (2) as follows:

- **(L1)**: If an individual network edge weight $Q_{\text{sgn},a}(j, k) > 0$, then the associated type I interaction effect $\beta_{jk}$ is non-zero in the model (2).

- **(L2)**: If an individual network edge weight $Q_{\text{sgn},a}(j, k) < 0$, then the associated type II interaction effect $\alpha_{jk}$ is non-zero in the model (2).

- **(L3)**: If an individual network edge weight $Q_{\text{sgn},a}(j, k) = 0$, then both $\alpha_{jk}$ and $\beta_{jk}$ are zero in the model (2).

The R-code for this whole procedure and a step-by-step guidance for its usage is available in S2 Appendix (See also Fig 2). In particular, the only parameter (by default) left for users to be specified case-specifically is the threshold parameter $r$. This is considered later in more detail.

## Simulation studies

The proposed method is evaluated and compared to the exhaustive search and GGM based approaches through simulated scenarios. These examples are based on data provided by DREAM9-challenge (organized in June 2014). This dataset [28] consists of 191 patients diagnosed with acute myeloid leukemia (AML) with measured expression levels of 231 proteins and phosphoproteins probed by reverse-phase protein array analysis each of which is following a standard normal distribution. New phenotypes are simulated conditionally on the expression levels of these proteins to have a known phenotype regulation mechanism and realistic dependencies between protein expression levels. We will use numeric subindexes $1, \ldots, 231$ to indicate a specific protein in the DREAM-challenge dataset starting from the ACTB.

**Model without the main effects.** The simplest part of our simulation model contains arbitrarily chosen six type I interaction terms controlling the variation of a normally distributed phenotype $Y$ without any main effects. Also, more complex interactions of two different types are incorporated into the model. First, we add two rectified linear unit (ReLU) terms $\Delta_{jk}(X_{ij}, X_{ik})$, defined to be equal to $X_j X_k$ if $X_j X_k \geq$ median$(X_j X_k)$ and zero otherwise. At this point, the simulation model is of the form:

$$\text{Model A}: \quad Y_A = X_{75}X_{150} + X_{100}X_{200} + X_{125}X_{215} + X_{25}X_{52}$$
$$+ X_{33}X_{66} + X_{88}X_{144} + \Delta(X_{12}, X_{183}) + \Delta(X_{109}, X_{54}) + \varepsilon.$$

Ten replicates of the phenotype vector were simulated based on the above model with the population intercept of zero. The common residual variance $\sigma_\varepsilon^2$ was fixed to $1.75^2$ for independent and normally distributed (zero-centered) residual terms $\varepsilon_1, \ldots, \varepsilon_{191}$ for each individual (the simulated replicates are available at the Additional file 2).

We also use another interaction function $\mathcal{C}_{+/-}(X_j, X_k)$ to mimic disrupted interactions in pathways among individuals with high or low phenotype values. These type of interactions are simulated with respect to the variables $X_j$ and $X_k$ backward as follows: We first compute the phenotype values $Y = (Y_1, \ldots, Y_{191})$ for each individual based on the simulation model A. For individuals with the phenotypic values larger than the 4/5 empirical quantile of $Y$, we overwrite the expression values of the protein $X_j$ as a function of the protein $X_k$ to induce correlation between them such that

$$X_{ik} = \pm X_{ij} + \varepsilon_i^*, \quad \text{where } \varepsilon_i^* \sim \mathcal{N}(0, 0.25^2), \tag{4}$$

only if $Y_i \geq Q_Y(2/3)$ and kept as original otherwise. In other words, such strong correlations are present only among individuals with high phenotype values. A subindex in $\mathcal{C}_{+/-}(X_j, X_k)$ denotes whether the induced correlation is positive or negative with reference to the symbol $\pm$ in the Eq (4). We used this rule to generate the following interaction set over individuals with phenotype values larger than the 4/5 empirical quantile of the phenotype vector resulted from the simulation model A:

$$\{\mathcal{C}_-(X_2, X_{170}), \mathcal{C}_-(X_{50}, X_{115}), \mathcal{C}_-(X_{44}, X_{99}), \mathcal{C}_+(X_{12}, X_{180}), \mathcal{C}_+(X_{60}, X_{125}), \mathcal{C}_+(X_{211}, X_{222})\}.$$

Furthermore, we overwrite the expression values of the proteins $X_{125}$ and $X_{75}$ as a function of the proteins $X_{215}$ and $X_{75}$ to induce strong correlation between them before the phenotypic truncation such that

$$X_{125} = X_{215} + \varepsilon, \quad \text{where } \varepsilon \sim \mathcal{N}(0, 0.25^2),$$

$$X_{75} = X_{150} + \varepsilon, \quad \text{where } \varepsilon \sim \mathcal{N}(0, 0.25^2).$$

The magnitudes of the induced correlations were approximately 0.95. These mimic the problematic collinearities between genes the interaction of which are important with respect to the phenotype.

**Applied methods**. To evaluate the efficiency of the proposed dependency metric, we begin by applying only the non-residual adjusted dPCCN approach: high and low groups are defined with respect to the empirical median value of the original phenotype $Y$. As a comparison, we estimate the sign-adjusted differential correlation co-expression networks (dCCN) in each scenario also by using the simple correlation matrices instead of the proposed part-correlation matrices. The same high and low groups are also used in the GGM based approach: The fused graphical LASSO algorithm via the JGL R-package [12] is used to estimate the high and low GGM networks and we refer to their difference as a differential GGM network (see [12] for details and S1 Appendix for the tuning parameter selection). In the exhaustive search approach, the interaction effects are estimated with the LASSO estimator [29] in which the penalty parameter is chosen by the cross-validation criteria. As will be explained in the materials and methods section, type I and II interactions are separated from each other by evaluating from data whether or not $|\Phi_{j,k}| = 0$ by using a user-specific threshold $r > 0$. Here we apply a relatively small threshold $r = 0.1$ such that $|\Phi_{j,k}|$ is deemed to be zero if $|\Sigma_{j,k}| < r$.

**Benchmarking**. The evaluation and comparison are done by using the receiver operating characteristics (ROC) curves [30]. The decision threshold value $a$ is shifted over the range of estimated network elements (or coefficient vector elements in exhaustive search) to produce the true positive rate (TPR) and false positive rate (FPR) for each value of $a$ such that

$$\text{TPR} = \frac{\text{The number of true positives}}{\text{The number of positives}} \text{ and } \text{FPR} = \frac{\text{The number of true negatives}}{\text{The number of negatives}}.$$

Truncated (at 0.2 FPRs) and non-truncated areas under the ROC curves (AUCs) averaged over the simulated replicates are displayed in Table 1 for each method (Model A columns).

**Results**. The most worrisome part of these results is a poor performance of the differential GGM approach which deserves to be noted given its popularity and trending usage in gene-gene interaction search—see e.g. [31]. The averaged non-truncated AUC is only 0.709. We like to highlight that interpreting results in terms of typical parametric forms should be done with caution when the differential GGM approach is applied. We argue that this problem is due to the conditioning property, which in fact, is the main reason for GGMs' popularity. A supportive example is given in S1 Appendix implying that their role in the differential network estimation scheme should be characterized more specifically. Another concern is that the exhaustive search approach does not have desired efficiency even though the model is relatively simple. The averaged AUCs and truncated AUCs were 0.761 and 0.713.

Let us now consider the proposed sign-adjusted dPCCN. The improvement in performance is tremendous in comparison to the reference methods—the non-truncated and truncated

**Table 1. Simulation studies.** Averaged areas under the truncated and non-truncated ROC curves (AUCs) over ten replicates in the simulated scenarios without simulated main effects (Model A) and with additional main effects (Model B). These datasets are analyzed using the proposed dPCCN procedure and dCCN method as well as the exhaustive search and GGM model based approaches as reference methods.

| Method | A: AUC (0.2 FPR) | A: AUC (1.0 FPR) | B: AUC (0.2 FPR) | B: AUC (1.0 FPR) |
|---|---|---|---|---|
| Sign dPCCN | **0.869** | **0.915** | **0.793** | **0.835** |
| Sign dCCN | 0.631 | 0.600 | 0.534 | 0.500 |
| Exh. search | 0.713 | 0.761 | 0.537 | 0.597 |
| dGGM | 0.707 | 0.709 | 0.618 | 0.620 |

AUCs were 0.915 and 0.869. Firstly, this indicates that traditional interaction search approaches suffer for more fundamental limitations than the lack of power and scalability that are often the subjects of interest. In addition to a more flexible model in the parametric sense, the use of tailored metrics to account for the inherent characteristics of gene-phenotype regulation mechanisms is clearly of high importance. For instance, the sign-adjusted dCCN is considerably less efficient in the above examples than the sign-adjusted dPCCN as expected. Namely, the non-truncated AUCs for the sign-adjusted dCCN and dPCCN were 0.631 and 0.915. Of course, the sign-adjusted dCCN, by the definition would be more efficient for identifying if a standard correlation coefficient is zero in the low group and non-zero in the high group, or vice versa. However, it becomes inefficient for more challenging scenarios involving, e.g., complex activation/deactivation patterns between genes. Thus, the proposed part-correlation metric is an indispensable additional element if we want to identify interactions in the estimation process.

**Model with the main effects.**   In this example, the phenotype replicates are simulated otherwise in the same way as in the previous example (using different replicates) but we also incorporate six strong main effects into the model i.e.,

$$\text{Model B}: Y_B = Y_A + \sum_{j=1}^{p} X_{ij}\beta_j + \varepsilon.$$

Here $\beta_j = 2$ if $j \in \{10, 30, 50, 70, 90, 100\}$ and zero otherwise. In this case, we apply the residual adjusted version of the sign-adjusted dPCCN method to account for the main effects. To estimate the main effects, we used the elastic net estimator with $\alpha = 1/3$ (a default value in the provided R-code) using the cross-validation based selection of the penalty parameter $\lambda$. Then, high and low groups are defined with respect to the median of the estimated residual vector.

**Results**. The results of this analysis are also displayed in Table 1 (Model B columns). It appears that the exhaustive search with truncated AUC of 0.537 and non-truncated AUC of 0.597 becomes unusable in this type of, fairly realistic, scenarios while the proposed residual- and sign-adjusted dPCCN method remains remarkably efficient with truncated AUC of 0.793 and non-truncated AUC of 0.835 despite the simulated main effects. The differential GGM approach is only slightly better than the exhaustive search—truncated AUC is 0.618 and non-truncated AUC is 0.620. These simulated scenarios reveal huge benefits that could be achieved with the proposed hybrid approach in comparison to the reference methods. This type of hybrid approaches could truly open new avenues for interaction search and, as an additional proof of concept, we illustrate its usage in a real acute myeloid leukemia example in the next section.

## Real data analysis—Acute myeloid leukemia

Acute myeloid leukemia (AML) is a hematological cancer of the myeloid line of blood cells and the prognosis of this disease is poor with an extremely low 5-year survival rate [28]. The recent advancements in high-throughput technologies have contributed to progress in leukemia research and especially to predict survival times from gene expression profiles [32–34]. However, the majority of these studies have focused only on single genes or their additive effects [4]. Although genetic interactions may help us better understand cancer biology and the development of new therapeutic approaches [35], the effects of gene-by-gene interaction on the survival times are not well known in cancers.

**Survival time analysis.**   We apply the proposed framework for searching prognostically valuable type I and II gene-by-gene interactions in AML. The same DREAM9 protein expression dataset is used as in the simulated examples but now the response variable represents real

patients' survival times after diagnosis. Out of these 191 patients, 142 died during the follow up with 77 weeks median survival time (quartile interval: [25, 103]).

We chose this survival analysis example to show that this method provides reasonable results even with problematic/incomplete datasets. In this case, difficulties arise from the most frequently appearing aspect of survival analysis, the right-censoring. This means that a patient has left the study before death, i.e. $C_i < Y_i$, where $C_i$ denotes the time of censoring and $Y_i$ is the actual survival time for an individual $i$ [36]. There are 49 censored observations in the DREAM9 dataset the most of which are above the median survival time (74 weeks) measured from non-censored observations.

We must note that revealing the most hidden and important features of this data is not the primary task here, and we are fully aware that some aspects might seem unreasonable from that perspective. This example serves to provide one kind of practical example of cross-exploiting the strengths of individual schemes to overcome the deficiencies of one and another. Therefore, we chose to use intentionally a particularly problematic dataset, which also might seem controversial on occasion.

In this case, for instance, we have to remove the censored observations for the network construction step. This clearly induces bias to the results [36] if one aims to estimate the exact effects of certain covariates on survival time. However, in this step, we aim to identify the most important interaction terms associated with the survival time. As we are interest on the effects sizes only on a relative level, we could tolerate a much higher amount of bias if we can ensure that the overall procedure is conservative enough. This is done by switching back to the parametric models, in which case the censored observations can be accounted for without technical problems, e.g., via the log-rank test [36]. Thus, this validation step(s) is used to filter out the false positives findings caused by the induced bias in the network estimation step.

In this example, we only apply the residual- and sign-adjusted version dPCCN method. The main effects are estimated with the elastic net estimator [37] using $\alpha = 1/3$ to obtain the estimated residual vector with the penalty parameter λ chosen by the cross-validation ($\lambda \approx 0.403$). Individuals are then divided into the high and low groups based on the median value of this estimated residual vector (71 observations in each group). As explained in the materials and methods section, type I and II interactions are separated from each other by evaluating from data whether or not $|\Phi_{j,k}| = 0$. To that end, we apply a relatively small threshold $r = 0.1$ such that $|\Phi_{j,k}|$ is deemed to be zero if $|\Sigma_{j,k}| < r$.

The estimated differential network structures are displayed in Fig 3 where the interaction types I and II are separated by green (type I) and red (type II) edges. Since the method itself is unpenalized, we used a hard-thresholding procedure to produce sparsity into the resulting differential network. For simplicity, the threshold value was chosen such that the number of network edges is less than 70 for both interaction types. Yet, how to define the most optimal threshold value is beyond the scope of this work.

**Type I interactions.** To show the explicit connection between the proposed framework and an exhaustive search we parametrically test all genes connected with green edges. This is done in accordance to the Aiken-West test [38]: A gene pair $(X_k, X_l)$ is tested by regressing the survival time $Y$ on both individual genes and their interaction term, i.e., $Y = X_k \beta_k + X_l \beta_l + X_k X_l \beta_{kl}$ over non-censored observations. At first, when testing the null-hypothesis of zero regression coefficients, the p-value associated with the interaction term should be relatively small. Then, the interaction term is deemed relevant if the corresponding p-value is smaller than the p-values associated with the main effects $\beta_k$ and $\beta_l$.

Now 56 out of 67 interactions ($\approx 84\%$) were also "positive findings" in terms of the Aiken-West test. Note that we used this test to only illustrate the connection between the proposed framework and an exhaustive interaction search. However, only two interesting type I

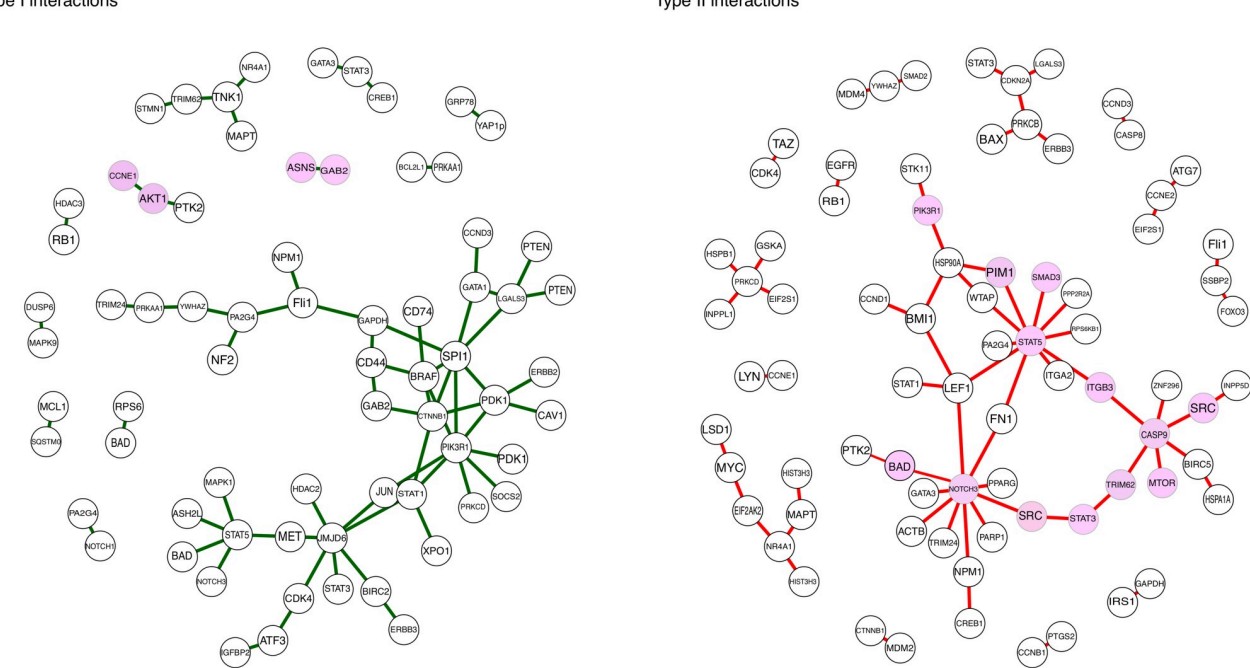

**Fig 3. Estimated differential networks.** Estimated differential networks with the proposed residual- and sign-adjusted dPCCN approach in the DREAM9-challenge protein expression dataset using the patients' non-censored survival times (142 observations) as response. The interaction types I and II are separated by green (type I) and red (type II) edges (plotted separately). A hard-thresholding was used to provide sparsity into the network structures such that the number of network edges is less than 70 for both interaction types. The estimated network structures are displayed only for connected nodes (with MiMI names) and the highlighted nodes indicate which network structures are discussed in detail.

interactions (highlighted in Fig 3) are considered in more detail as an example: Interactions between RAC-alpha serine/threonine-protein kinase (AKT1) and cyclin E1 (CCNE1) as well as between asparagine synthetase (ASNS) and the antibody phospho-gab2 (Tyr452) of GRB2-associated-binding protein 2 (GAB2). The Aiken-West test results for these findings are given in S1 Table.

**Validation in an independent TCGA dataset.** The validation of these two interaction terms is based on the current literature and survival analysis performed in an another independent AML cohort. This cohort includes RNA-sequencing for 173 AML patients with measured expression levels of around 20 000 genes provided by The Cancer Genome Atlas (TCGA; LAML data available at https://cancergenome.nih.gov/). RNA and protein expression levels are expected to have a high correlation, but significant variation in correlation among genes [39]. Thus, the validation of interaction terms derived from protein expression data of DREAM9 in RNA expression level of TCGA is conservative, and suggests that the same interactions are discovered using either one. However, the lack of congruence would not necessarily invalidate the results derived from the other type of dataset due to different biological control mechanisms of RNA and protein level expression.

To evaluate the prognostic power of these findings, all patients in TCGA dataset (including censored observations) are classified into distinct gene-expression profile based risk-groups as follows: If the expression value of the interaction term (CCNE1, AKT1) is below its $q$th quantile, the patient is classified into the low-risk group (low-expression) and otherwise into the high-risk group (high-expression). Note that the effects of gene-by-gene interactions (CCNE1, AKT1) and (ASNS, GAB2.pY452) are of opposite signs. Thus, if the value of the interaction

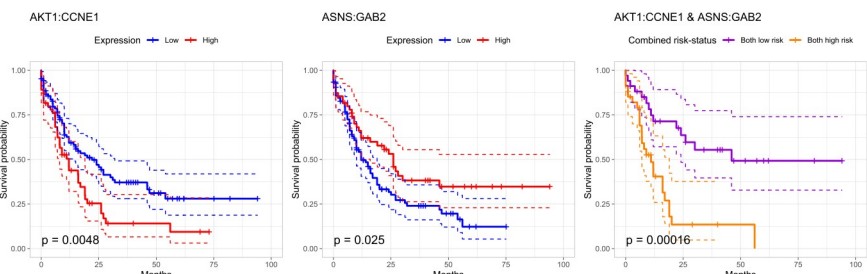

**Fig 4. Survival analysis using type I interactions.** Kaplan-Meier curves and the 95% confidence intervals for three different classifiers in the TCGA dataset (173 observations). The first two panels represent the (AKT1, CCNE1) interaction term based classifier and the (ASNS, GAB2) interaction term based classifier. Red and blue survival curves correspond to high and low expression values. The last panel is a combined classifier; the low-risk group patients based on both interaction terms (AKT1, CCNE1) and (ASNS, GAB2) are classified into the combined low-risk group (purple curve) and into the combined high-risk group (orange curve) with the same logic. The p-values of the corresponding log-rank tests are also reported.

term between (ASNS, GAB2.pY452) is above its $(1 − q)$th quantile, the patient is classified into the low-risk group (high-expression) and otherwise into the high-risk group (low-expression). We chose $q$ to be 2/3 in order to emphasize the high-risk profiles. These gene-expression profiles are then analyzed via Kaplan-Meier analysis (Fig 4).

**AKT1 and CCNE1**. Constitutive phosphoinositide 3-kinase (PI3K) and AKT signaling are repeatedly reported in AML studies [40]. However, there is considerable variation in the effect of these pathway inhibitors among AML patients [41]. However, it might be that the prognostic importance of AKT1 becomes crucial through the interaction with CCNE1 (Fig 4, left panel) which plays a key role in cell proliferation [42]. Even though few studies [43, 44] has mentioned the possible prognostic value of the expression of CCNE1, the accurate prognostic role regarding CCNE1 has remained unclear. However, once we consider the interaction term (AKT1 and CCNE1), high expression values (red curve) reduce median survival times clearly in the TCGA dataset (Fig 4): Median survival times were 22 months in the low-risk group and 12 months in the high-risk group, respectively.

**ASNS and GAB2**. When the same expression profile based risk classification is done based on the interaction term between ASNS and GAB2, median survival times were 26 months in the low-risk group (high expression) and 12 months in the high-risk (low expression) group using the TCGA dataset. However, it has been shown in [45] that high ASNS expression values reduce survival times. In light of this example, it appears that we should be careful with such a conclusion. Namely, already in the Aiken-West test performed in the DREAM9 dataset, we get an indication that its prognostic importance in AML may be in its interactive nature with GAB2 rather than as an independent prognostic factor (see S1 Table).

**Combined risk classifier**. We also illustrate how accurately these two interaction terms together classify patients into high- and low-risk groups. Patients that are in the low-risk group based on both interaction terms (AKT1, CCNE1) and (ASNS, GAB2) are classified into the combined low-risk group. Respectively, patients that are in the high-risk group based on both interaction terms are classified into the combined high-risk group. Other individuals are removed. This classifier has remarkable accuracy even though it is based only on two interaction terms. In the combined high-risk group (orange curve in Fig 4, right panel), the median survival time is only 11 months in contrast to 46 months in the combined low-risk group (purple curve in Fig 4, right panel). Nearly three years difference in median survival times between

the high- and low-risk groups (p-value ≈ 0.00016) indicates that these findings have prognostic value in AML.

**Type II interactions.** Type II interactions in Fig 3 cannot be tested explicitly using the same parametric interaction test similar to the type I interactions. In general, the effects of these type II interactions on a patient's outcome can be relatively complex. Therefore, their proper validation should be done always from the biological point of view which is beyond the scope of this paper. Thus, we will rely on the current literature and select a few representative examples that happen to share the same parametric form by which they can be used to separate individuals into different risk groups.

**Validation.** It was interesting that the major parts of the signaling pathways known to have a major impact on AML progression [46–49] are overlapping with the estimated network structures. Based on these previously reported AML-related pathways, we select a smaller representative subnetwork to be analyzed in more detail (highlighted in Fig 3).

Signal transducers and activators of transcription STAT3 and STAT5 are both downstream effectors of several tyrosine kinase oncogenes including proto-oncogene tyrosine-protein kinase SRC each of which are central nodes in the estimated network [50]. Interestingly, a particular kinase inhibitor drug based treatment (sorafenib) for AML has been shown [51] to block SRC kinase-mediated STAT3 phosphorylation. The STAT5 activation, on the other hand, is regulated by an interplay between SRC family kinases and the mammalian target of rapamycin (MTOR) via AKT/MTOR signaling pathway [52]. These terms are closely related in the estimated network through caspase-9 (CASP9) and integrin beta-3 (ITGB3). Moreover, we like to bring forth that both ITGB3 and tripartite motif containing 62 (TRIM62) are binding the densest clusters together in the estimated network. This is particularly interesting due to their possible interplay [53].

Further, STAT5 can interact with mediators of the PI3K/AKT signaling cascade which plays a central role in the cancer cell survival [54]. This might explain other connections in the estimated network. For instance, the downstream targets of the signaling pathway PI3K-AKT include the BCL2-associated agonist of cell death (BAD) and CASP9 [55]. Furthermore, AML-specific down-regulation of BAD/BCL2 plays a critical role in NOTCH-mediated apoptosis in AML [56]. In our study, phosphorylated BAD (pS112) were associated with neurogenic locus notch homolog protein 3 (NOTCH3) in the estimated network. The role of entire NOTCH family, including NOTCH3, in AML is not well-understood and there have been conflicting studies about its role in AML [56]. However, since the neighborhood of NOTCH3 appears to be quite dense, it would be worth for further consideration.

We proceed by considering two STAT5 related examples more closely. To that end, we use RNA sequencing expression data from the TCGA database via GEPIA [57] to illustrate the prognostic power of these selected/representative pairs in an independent dataset.

**Signal transducer and activator of transcription 3 and 5.** STAT3 and STAT5 are both members of the STAT protein family. STAT5 is consisting of two related proteins, STAT5A and STAT5B, that share about 90% identity at the amino acid level [58]. As they are separated in the validation dataset but not in the DREAM9 dataset, we analyze and validate all identified interactions with respect to STAT5A and STAT5B separately.

**STAT5 and SMAD3**. SMAD family member 3 (SMAD3) is one of the receptor-regulated effector proteins (R-Smads) in the transforming growth factor beta (TGF$\beta$) signaling pathway [59]. In particular, it is shown that ligand-induced activation of SMAD3 by the protein complex activin and TGF$\beta$ leads to a direct inhibition of STAT5 transactivation [60]. Our results suggest that this interplay between STAT5 and SMAD3 contains highly valuable prognostic information. Using 20% and 80% cut-off points to classify high- and low expression ratio groups in GEPIA, high STAT5/SMAD3 TPM ratios (Transcripts Per Million) were associated

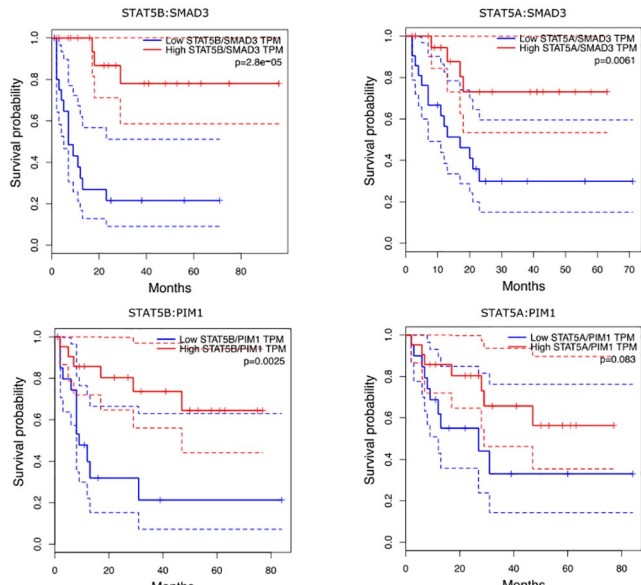

**Fig 5. Survival analysis using type II interactions.** Results from survival analysis including survival curves (and 95% confidence intervals) and statistics using GEPIA software with 20% and 80% cut-off points to classify high- and low expression ratio groups (21 individuals in both groups). In each panel, p-values of the log-rank tests are reported for hypothesis tests of no differences between groups.

with more severe prognosis than low STAT5/SMAD3 TPM ratios with the p-value of $2.8 \cdot 10^{-5}$ for STAT5B and $6.1 \cdot 10^{-3}$ for STAT5A in the log-rank test (Fig 5). It also seems that this interaction term could not been found by typical parametric interaction tests. The parametric Aiken-West test in the DREAM9 dataset gave a p-value of 0.476 when testing the null-hypothesis of zero-valued effects.

**STAT5 and PIM1**. Expression levels of proto-oncogene serine/threonine-protein kinase (PIM1) is known to be regulated by the JAK/STAT pathway involving both STAT3 and STAT5 proteins [61]. When the interaction terms between PIM1 and STAT5 are tested with GEPIA, high PIM1/STAT5B TPM ratios levels were more common with short survivals than low PIM1/STAT5B TPM levels with p-value of 0.0025 in the log-rank test. Respectively, individuals with high PIM1/STAT5A TPM ratios had shorter survival times than with low PIM1/STAT5A TPM ratios with the p-value of 0.083 in the log-rank test. Also in this case the DREAM9 dataset was not conflicting with the null-hypothesis of zero-valued interaction effect in the Aiken-West test (p-value of 0.326).

## Discussion

The examples of this paper reveal a remarkable advance of the proposed approach over the commonly used reference methods. However, even more important is to observe that the traditional exhaustive search approach and popularity gained GGM based differential networks show a considerable lack of efficiency in the presence of typical attributes of gene/protein expression data. In particular, the representative real prognostic analysis showed how important interaction types might remain unidentified due to the limitations of common approaches. The message of these results is therefore dual; we shared our observation about the weaknesses of the mainstream methods but also provide a tailored alternative that has a huge potential to open new avenues for interaction search with a significant impact on many

important fields like prognostic analysis. Due to the method's ease of usage and flexibility for a variety of biological data, we foresee the wide applicability of this method with immediate practical relevance.

However, the proposed method is based on an explicit link between the elements of differential co-expression networks and parametric interaction models (presented in the materials and methods section) only of the form (2) and relies heavily on zero-centered scaling of the explanatory variables. By using this same framework, it would be interesting to see how a different kind of scaling of the explanatory variables would reflect the form of the underlying model. Especially, it is an open question whether some scaling of the explanatory variables would enable higher than second-order type I interactions to be identified in this framework and what kind of dependency metric (in place of the proposed part-correlation metric) is required for such purpose. These issues are beyond the scope of this paper and left for future studies.

It is also noteworthy that we based the proposed approach on sample correlation/covariance type matrices instead of popularity gained inverse correlation/covariance matrices i.e. GGMs. However, this should not be considered as criticism against GGMs since they can obviously recover the co-expression networks much more efficiently than dCCNs exactly due to the ability to distinguish these direct dependency from indirect ones. This is rather a matter of purpose—the aim in the common differential network studies is to recover the gene co-expression patterns in two or more classes and compare the overall network dynamics in a more causal sense between them. Here the focus is on mapping the parametric interaction model into the pairwise dependencies between genes regardless of their conditional dependency structure.

Moreover, the estimation of the inverse covariance matrix $\Sigma^{-1}$ via maximum likelihood is not possible when the empirical correlation matrix $S$ is singular (for example when $n < p$ since rank$(S) \leq n - 1$). Some penalized estimators have been made for estimating the inverse covariance matrix $\Sigma^{-1}$ in the high-dimensional settings enabling non-singular, and even sparse results. For example, applying the LASSO-penalty to the elements of $\Theta = \Sigma^{-1}$ leads to convex optimization problem proposed in the paper of [62] enabling the inverse covariance matrix to be estimated even if $n < p$. However, this increases computational complexity which is between $\mathcal{O}(p^3)$ and $\mathcal{O}(p^4)$ for a row-by-row block coordinate method. Thus, one benefit of using correlation co-expression networks instead of GGM based alternatives is their extremely low computational complexity of learning.

## Methods and models

Here we provide a synergistic framework based on parallel consideration of gene-by-gene interaction models and novel quantitative trait-specific co-expression networks. In particular, we address the following issues: (1) What is the explicit link between the co-expression networks and different types of parametric interaction terms? (2) How this link can be used to derive an efficient and flexible trait-specific co-expression network estimation metric? (3) How do we properly account for the inherent characteristics of gene-to-phenotype architectures (e.g. strong main effects and collinearities between genes) in our network construction?

### Co-expression network estimation schemes

There exist two major paradigms for constructing linear co-expression networks based on unconditional (indirect) and conditional (direct) dependencies. The co-expression networks representing unconditional relationships between genes are often defined by the covariance (or correlation) matrix $\Sigma$. Such networks are referred as covariance/correlation co-expression

networks (CCNs). For conditional co-expression network estimation, a popular option is to use a inverse covariance/correlation matrix $\Sigma^{-1}$ which is known and later referred as GGMs. It has been shown that under the assumption of normality an element of $\Sigma^{-1}$, say $\Sigma_{j,k}^{-1}$, is zero, if and only if genes $X_j$ and $X_k$ are conditionally independent given the rest of the genes [25]. However, we have shown that GGM based methods lack of efficiency for our purposes exactly due to the conditioning property they are favored in the first place (see S1 Appendix). Thus, our method is formulated only for the correlation/covariance co-expression networks.

### Trait specific co-expression networks

Let us assume that the phenotype $Y$ is regulated by the interaction model (2). We begin by defining high and low groups of individuals based on the observed values $Y_1, \ldots, Y_n$ of the quantitative phenotype $Y$ such that:

- A **high-group** is consisting of individuals, whose phenotype values $Y_i$ fall into a critical region, defined to be the top $(1-a) \times 100\%$ $(a \in {]}0, 0.5])$ highest values of phenotype among all individuals, i.e., $Y_i \geq Q_Y(1-a)$, where $Q_Y(\cdot)$ is a quantile function of the phenotype distribution.

- For **a low-group**, a critical region corresponds the $a \times 100\%$ lowest phenotype values i.e. a control group is consisting of individuals to whom $Y_i \leq Q_Y(a)$ holds.

The high and low groups are thereby conditioned to the corresponding tails of the phenotype distribution such that the magnitude of this conditioning depends on the $a$ value. Note that unless $a$ equals to 0.5, individuals/samples between high and low groups are omitted from the analysis so we prefer that $a = 0.5$. Let us proceed by introducing so-called truncated network structures corresponding to the high and low groups:

- A **high network** $G^{1-a} = \Sigma^{1-a}$ is constructed over individuals in a high group, i.e., to whom $Y_i \geq Q_Y(1-a)$.

- A **low network** $G^a = \Sigma^a$ is estimated over individuals, to whom $Y_i \leq Q_Y(a)$ holds, respectively.

However, we are not interested in the high and low networks as such but rather on the dissimilarities between them as in [11, 12]. We therefore define a differential correlation co-expression network (dCCN) structure as $C_a = |\Sigma^{1-a} - \Sigma^a|$ where the absolute value is taken over the covariance/correlation matrix element-wise. The main idea will be first illustrated with additional naive assumptions regarding the underlying model (2). When these assumptions are relaxed towards more realistic scenarios, new metrics are derived to modify data such that these naive assumptions hold again.

### Identifying interactions of type I and II

Determining a one-to-one correspondence between the type I interactions in the model (2) and the differential networks is based on the following observation. Under certain conditions, the dependencies in the high and low groups between two genes $X_j$ and $X_k$ are of different signs with equal magnitudes if they form an important type I interaction term $X_j X_k$ in the model (2). Conversely, if $X_j X_k$ is not an important interaction term of type I or type II, the dependencies between $X_j$ and $X_k$ are equal to each other in the low and high groups. This implies that there exists an edge between genes $X_j$ and $X_k$ (assuming that $\alpha_{jk} = 0$) in the corresponding differential network only if the term $X_j X_k$ is relevant type I interaction in the model (2) (see the example (1) in S1 Appendix).

Respectively, correlation/covariance dependency-metric (CCN) specific type II interactions of the model (2) are identified by distinguishing certain dependency patterns between genes in the high and low groups (see also the example (2) in S1 Appendix) which is self-evident by their definition. The following proposition (proposition 1—proved in S1 Appendix) characterizes these statements more precisely.

**Proposition 1** *Let $\Sigma_{j,k}^{a}$ and $\Sigma_{j,k}^{1-a}$ denote the correlations between variables $X_j$ and $X_k$ in the low and high groups. Let us use a specifying notation $\alpha_{jk}^{*}$ for the type II interaction effect of genes $X_j$ and $X_k$ that contribute to the phenotypic variation through a linear co-expression relationship which is associated only with high (or low) phenotypic values. We also assume that $X_k$ and $X_j$ are independent before the phenotypic truncation (Assumption A) and that the main effects $\beta_k$ and $\beta_j$ are zero in the model (2) (Assumption B). Then we have that*:

- *Invariant property*: *If the interaction term $X_j X_k$ and the response variable $Y$ are independent then $C_a = |\Sigma_{j,k}^{a} - \Sigma_{j,k}^{1-a}| = 0$.*

- *Property A1*: *If the interaction effect $\beta_{jk}$ in the model (2) is non-zero, and all type II interactions among genes $X_j$ and $X_k$ are zero, then $\Sigma_{j,k}^{a} = -\Sigma_{j,k}^{1-a} \neq 0$.*

- *Property B1*: *If the effect $\beta_{jk}$ in the model (2) is zero and $\alpha_{jk}^{*} > 0$ then $|\Sigma_{j,k}^{a}| > 0$ and $\Sigma_{j,k}^{1-a} = 0$ or vice versa if $\alpha_{jk}^{*} < 0$.*

This proposition forms the basis for the explicit link between the model (2) and a new kind of differential network estimation method introduced in the next section.

## Sign-adjusted estimation

Here we present a sign-adjusted dCCN which exploits the properties A1 and B1 in the proposition (1) to categorize interaction types I and II in the estimation process based on the standard correlation/covariance dependency metric. Let us consider a dCCN $C_a = |\Sigma^{1-a} - \Sigma^{a}|$ where the truncated correlation matrices $\Sigma^{a}$ and $\Sigma^{1-a}$ represent the low and high networks for some truncation point $a \in ]0, 0.5]$ in relation to the model (2). Then the sign-adjusted dCCN $C_{\text{sgn},a}$ is defined as $C_{\text{sgn},a} = C_a \odot \Pi(\Sigma^{1-a}, \Sigma^{a})$ where $\odot$ is an element-wise matrix multiplication operator. The element-wise function $\Pi(\cdot, \cdot)$ is defined such that

$$
\Pi(\Sigma_{j,k}^{1-a}, \Sigma_{j,k}^{a}) = \begin{cases} 1 \text{ if } \text{sgn}(\Sigma_{j,k}^{a}) = -\text{sgn}(\Sigma_{j,k}^{1-a}), \\[2mm] -1 \text{ if } |\Sigma_{j,k}^{a}| > 0 \text{ and } |\Sigma_{j,k}^{1-a}| = 0, \\[2mm] -1 \text{ if } |\Sigma_{j,k}^{1-a}| > 0 \text{ and } |\Sigma_{j,k}^{a}| = 0, \\[2mm] 0 \text{ if } |\Sigma_{j,k}^{a}| = 0 \text{ and } |\Sigma_{j,k}^{1-a}| = 0. \end{cases} \tag{5}
$$

The $(j, k)$th element of the sign-adjusted dCCN is denoted by $C_{\text{sgn},a}(j, k)$. To evaluate from data whether or not $|\Sigma_{j,k}| = 0$, we can perform a simple hypothesis test with an appropriate Bonferroni correction [63] to account for the multiple testing problem. Alternatively, we could simply apply a relatively small threshold $r > 0$ such that $|\Sigma_{j,k}|$ is deemed to be zero if $|\Sigma_{j,k}| < r$ which is known as a hard-thresholding procedure [25].

While the regular dCCN (a simple difference of two correlation networks) captures the effects in spurious form, the sign-adjusted dCCN characterizes the link between the differential network elements and the effects in the model (2) as follows:

- **(L1)**: If $C_{\text{sgn},a}(j, k) > 0$ then the associated type I interaction effect $\beta_{jk}$ is non-zero in the model (2).

- **(L2)**: If $C_{\text{sgn},a}(j, k) < 0$ then the associated type II interaction effect $\alpha_{jk}^*$ is non-zero in the model (2).

- **(L3)**: If $C_{\text{sgn},a}(j, k) = 0$ then both $\alpha_{jk}^*$ and $\beta_{jk}$ are zero in the model (2).

Note that the links L1-L3, at this point, rely on the naive assumption of the propositions (1-2). Moreover, as dCCNs are based on standard correlation matrices, they are only capable of identifying if genes are linearly dependent in one group but not in another. In the next sections, we generalize these ideas to be suitable in more realistic and complex scenarios.

## Violation of the independence assumption

Let us consider the violation of the assumption (A) in the propositions (1-2). Differential co-expression network type approaches including [11, 12, 31] are poorly capable of finding interaction terms $X_k X_l$ when genes $X_k$ and $X_l$ are strongly correlated before the phenotypic truncation. This is due to fact that the high and low group construction cannot "break" the strong dependency between two genes and therefore the property A1 in the proposition (1) does not hold anymore as shown in S1 Appendix

Therefore, interaction terms are not identifiable if the correlation between genes is already strong before the phenotypic truncation. The dependencies between genes before truncation should be therefore accounted for by removing these dependencies in the estimation process.

## A novel part-correlation metric

We propose a so-called truncated part-correlation matrix $\Phi_{j,k}^a = \text{cor}(X_k^a, \varepsilon_{j|k}^a)$ to remove linear dependencies between genes $X_j$ and $X_k$ before the phenotypic truncation. Here $\varepsilon_{j|k}$ is the residual resulting from regressing $X_j$ against $X_k$ in the non-truncated data. Each matrix element $\Phi_{j,k}^{a/1-a}$ is then calculated as pairwise correlations between $X_j^{a/1-a}$ and $\varepsilon_{j|k}^{a/1-a}$ (or $X_k^{a/1-a}$ and $\varepsilon_{k|j}^{a/1-a}$) for all $1 \leq j, k \leq p$ ($a/1 - a$ refers to high and low groups simultaneously). We call these matrices as truncated part-correlation matrices to make clear difference to the partial correlation/covariance matrices in which the conditioning is performed over all genes excluding the pair $X_j$ and $X_k$.

This seems rather counter-intuitive at first since the correlation between two variables after removing the linear relationship between them is zero. However, now the assumption (A) in the proposition (1) holds again and the systematic dependency behaviour described in the property A1 remains between $X_j$ and $\varepsilon_{j|k}$ (or $X_k$ and $\varepsilon_{k|j}$) when constructing the high and low groups. Nevertheless, let $\Phi^a$ and $\Phi^{1-a}$ denote the truncated part-correlation matrices in the low and high groups for some truncation point $a \in \, ]0, 0.5]$. Then the part-correlation matrix based differential network $Q_a$ (dPCCN) and its sign-adjusted version $Q_{\text{sgn},a}$ with the truncation point $a \in \, ]0, 0.5]$ are defined as

$$Q_a = |\Phi^{1-a} - \Phi^a| \text{ and } Q_{\text{sgn},a} = Q_a \odot \Pi(\Phi^{1-a}, \Phi^a). \tag{6}$$

The $(j, k)$th elements of the dPCCN and sign-adjusted dPCCN are denoted by $Q_a(j, k)$ and $Q_{a,\text{sgn}}(j, k)$. The following properties (the proof of which is given in S1 Appendix) are analogous to the invariant and A1 properties in the proposition (1), except it only assumes that genes involved in any phenotypically important interaction term do not have significant main effects in the model (2).

**Proposition 2** *Let $\Phi_{j,k}^a$ and $\Phi_{j,k}^{1-a}$ denote the part-correlations between variables $X_j$ and $X_k$ in the low and high groups. Let us assume that the corresponding main effects $\beta_k$ and $\beta_j$ are zero in the model* (2). *Then we have that*:

- **Invariant property**: *If the interaction term $X_j X_k$ and the response variable $Y$ are independent then $\Phi_{j,k}^a = \Phi_{j,k}^{1-a} = 0$.*

- **Property C1**: *If $\beta_{jk} \neq 0$ and $\alpha_{jk} = 0$ in the model* (2) *then $\Phi_{j,k}^a = -\Phi_{j,k}^{1-a} \neq 0$.*

In comparison to the dCCN version, the dPCCN method accounts for more complex activation/deactivation patterns between genes. However, the parametric forms of this kind of type II interaction effects are more implicit (see S2 Appendix) than type II interaction effects $\alpha_{jk}^*$ in the property B1, and need to be specified always case-specifically. Now the statements L1-L3 hold for the sign-adjusted dPCCN elements $Q_{a,\mathrm{sgn}}$ even if the assumption (A) does not. However, so far the main effects of individual genes have not been accounted for. This significantly undermines the possibility for interaction terms to be found for several reasons discussed in the next section.

## Violation of the main effect assumption

Let us assume that some important type I interaction term $X_j X_k$ is consisting of genes with large main effects $\beta_j$ and $\beta_k$ in the model (2). In other words, the main effect assumption of the previous propositions does not hold. However, it is required for the property A1 (see proof for the proposition (1) in S1 Appendix) to be true that $E(X_j^a) = E(X_k^a) = E(X_j^{1-a}) = E(X_k^{1-a}) = 0$. However, genes with strong main effects are linearly related to the phenotype by which high and low groups are formed implying that this equality is not valid anymore. Another problem is that the proportion of the phenotypic variation explained by the gene-gene interaction might be insufficient for the interaction terms to be identifiable with small sample sizes [64–66].

**Residual step**. The parallel consideration of the underlying parametric model (2) enables an additional residual step to be used [64–66] by which the network estimation could be done independently on the main effects: The main effects $\beta_k$ ($k = 1, \ldots, p$) are first estimated in the model (1) without the interaction terms to get the residual estimates

$$\hat{\varepsilon}_i = Y_i - \hat{\mu} - \sum_{j=1}^{p} X_{ij}\hat{\beta}_j. \tag{7}$$

The estimated residual vector $\hat{\varepsilon}_i$ is independent of the main effects and considered as a new response variable for the interaction terms such that

$$\hat{\varepsilon}_i = \sum_{k>j} X_{ij}X_{ik}\beta_{jk} + \sum_{k>j}\alpha_{jk}g(X_{ij}, X_{ik}) + \varepsilon_i^*. \tag{8}$$

Here the random error $\varepsilon_{i,j}^*$ of order two is assumed to follow a normal distribution with a mean of zero and variance equal to $\sigma_*^2$. Now instead of dividing individuals into the high and low groups based on the original phenotype $Y_i$ we use the estimated residual values $\hat{\varepsilon}_i$. This yields that the main effect assumption in the previous propositions can be assumed if the estimated residual vector $\hat{\varepsilon}_i$ is used as a response variable. Networks estimated by using this residual step are referred as residual-adjusted networks.

## Supporting information

**S1 Appendix. Numerical examples, proofs and additional analyses.**
(PDF)

**S2 Appendix. R-codes and simulated data replicates.** See also a GitHub repository https://github.com/JAJKontio/model_diffnet.git.
(ZIP)

**S1 Table. Estimated effect sizes, standard errors and p-values for the main and interaction effects on survival times in the Aiken-West interaction test.** Estimation is done over patients with non-censored survival times (142 individuals) in the DREAM9 AML dataset.
(PDF)

## Author Contributions

**Conceptualization:** Juho A. J. Kontio, Mikko J. Sillanpää.

**Formal analysis:** Juho A. J. Kontio, Tanja Pyhäjärvi.

**Methodology:** Juho A. J. Kontio, Mikko J. Sillanpää.

**Software:** Juho A. J. Kontio.

**Visualization:** Juho A. J. Kontio.

**Writing – original draft:** Juho A. J. Kontio.

**Writing – review & editing:** Juho A. J. Kontio, Tanja Pyhäjärvi, Mikko J. Sillanpää.

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
