## [Decision Letter · Decision Letter 0]

29 Sep 2020

Dear Dr Sillanpää,

Thank you very much for submitting your manuscript "Model guided trait-specific co-expression network estimation as a new perspective for identifying molecular interactions and pathways" for consideration at PLOS Computational Biology.

As with all papers reviewed by the journal, your manuscript was reviewed by members of the editorial board and by several independent reviewers. In light of the reviews (below this email), we would like to invite the resubmission of a significantly-revised version that takes into account the reviewers' comments.

Please respond to all reviewer comments. Specifically, please consider major text edits, adding schematics of the methods used, and usable codes to improve clarity and reproducibility of the presented analyses.

We cannot make any decision about publication until we have seen the revised manuscript and your response to the reviewers' comments. Your revised manuscript is also likely to be sent to reviewers for further evaluation.

Sincerely,

Pejman Mohammadi, Ph.D.

Guest Editor

PLOS Computational Biology

Florian Markowetz

Deputy Editor

PLOS Computational Biology

Please respond to all reviewer comments. Specifically, please consider major text edits, adding schematics of the methods used, and usable codes to improve clarity and reproducibility of the presented analyses.

Reviewer's Responses to Questions

**Comments to the Authors:**

Reviewer #1: This manuscript proposes a new method for estimating co-expression networks to identify interactions between genes. Remarkable improvements of the new model-based method over alternatives are shown theoretically and in simulations, and the application is demonstrated with a widely known biomedical case example.

1. The manuscript is in general clearly written but not accessible to non-specialists due to many technical details of the method that lack intuitive explanation. I did not find obvious flaws in the methodology and derivations, and justification has been provided for individual steps in the algorithm. The overall description would benefit from a more intuitive explanation for instance in the form of an illustration. Some of the benchmarking details could be moved to supplementary material in order to improve readability, and the most important explanations for the improved performance of the proposed method could be more clearly stated in Discussion.

2. Observed co-expression provides evicence of association rather than interaction. Please reconsider the terminology or justify the use of the term "interactions".

3. The proposed method appears to perform well but the overall novelty is limited since a number of co-expression techniques have been introduced over the last 20 years, and although source codes are shared in the supplementary material, this work is not delivering a general-purpose ready-to-use implementation. This forms a major limitation for practical application and further benchmarking in other studies.

* Minor

l. 41 "A little" -> "Little"

l. 277: paper -> work?

l. 447: ".," -> ","

Table 1: the best performing methods (numeric values) could be highlighted (bolded) in order to facilitate interpretation

Reviewer #2: "Model guided trait-specific co-expression network estimation as a new perspective for

identifying molecular interactions and pathways" by Kontio, et al.

The authors present a method for interaction network inference, which combines features of co-expression networks and parametric interaction models. The idea is quite beautiful at its simplicity, which, however, is somewhat hard to grasp while reading the first parts of the manuscript. Therefore, some improvement in the flow of Abstract and Introduction Sections is warranted. Code availability should be guaranteed.

Major Comments:

Please explain more concretely already in the abstract, what the paper’s achievement is. Now, I felt that I had to read up to page 6-7 to get an understanding this. The description “framework for parallel consideration of parametric interaction models with quantitative traits and co-expression networks based on a previously uncharacterized link between them” should be made more tangible, in my opinion.

A schematic diagram of the approach could make the paper accessible to a wider audience. This could include a general illustration of: how differential co-expression networks work and how parametric interaction models work, what is the link between them and how your work utilizes this. The Objectives on page 6 are an excellent summary of the goal. If these came into the illustration as well, we would have an overview of the idea in the paper shown also to a wider audience.

Please go through the manuscript and reduce complexity in paragraphs, where there is a subordinate clause in the middle of a paragraph without it having been made distinct with punctuation. For example, “For instance, in transcriptional interactions where a transcription factor binds to promoter regions of a particular gene to regulate its expression levels can be disrupted in cancers” is hard to grasp without reading several times. At simplest, add helpful punctuation, as in “For instance, in transcriptional interactions*,* where a transcription factor binds to promoter regions of a particular gene to regulate its expression*,* levels can be disrupted in cancers”. Refactoring, though, would in most cases lead to a better result than this simplest suggestion. Especially the Introduction Section suffers from a high load of this type of complex paragraphs.

I did not find the S2 Appendix or “Data and software” Supplement, where the code was referred to be found. Code availability is key for a method’s impact. It would be highly beneficial, if the code was available in an accessible form, for instance, as an R/Bioconductor package, or at least on GitHub or similar repository. This would give the authors also a platform to making further work with the tool.

I am not convinced that it is appropriate to estimate the confidence intervals of a method with the ci.auc function of the pROC package. Should you not be bootstrapping the original data before model fitting instead of bootstrapping the predictions after model fitting? Won’t the latter approach lead to a great overconfidence as the bootstrapping takes place only after model fitting?

I do not understand, why the authors are omitting the right-censored observations from the study. As the authors describe, it is not conventional and it leads to a biased inference of the hazard function. Wouldn’t it have been just simpler and more appropriate for the paper to include all observations? Could you provide a sensitivity analysis regarding the omission of observations?

Why is elastic net estimator used in the real data experiment, whereas L1/LASSO is used in the simulated data experiments? The choice should be motivated. Furthermore, it would be beneficial to provide a way forward regarding which method should the reader choose, if one were to use you’re approach for new data.

Combined risk classifier: Why are only low- and high-risk patients included in the combined risk classifier, whereas all were included in the single-interaction classifiers? For consistency, result should be presented for complete data, too.

Minor Comments:

Abstract: Clarify paragraph: what is referred to by “them”?

“We provide a framework for parallel consideration of parametric

interaction models with quantitative traits and co-expression networks based on a

previously uncharacterized link between them.”

Figure 1: Colors in the figure are not friendly for color blind.

Results:

p.18,l.431: Wording with “consistently” is perhaps overstatement for a p-value of 0.083.

Thanks for an interesting idea and a nice work!

Reviewer #3: The authors present a new method to identify important gene-gene interactions with respect to the relationship between expression and certain phenotypes. Specifically, their model combines parametric interactions with those represented by co-expression networks. They show that this type of model uncovers biologically meaningful interactions and allows for better associations with phenotypes such as disease survival time.

The proposed method builds conceptually on existing methods in interesting ways, and seems to be a promising tool for biological discovery. My comments are mainly in regards to strengthening and clarifying the framing and validation of the method.

1. It seems that DREAM9 data analysis involved protein expression levels, while the TCGA/GTEx data analysis involved RNA expression levels. Since RNA and protein levels are affected differently by regulation, often do not correlate highly, and provide different information about biological state, this distinction should be discussed. Is this method better suited toward one or the other? Since interactions derived from protein levels were validated using RNA-seq expression levels, does that suggest that the same important interactions can be discovered using either one?

2. This varies from person to person and is not essential, but I would benefit from a figure containing a diagrammatic overview of the method. It could show what kind of data it uses, how it uses it to model interactions, what information it produces, and how that information could be applied to e.g. prognosis.

3. Validation of type II interactions was done using data from TCGA and GTEx projects via GEPIA. My understanding is that GTEx individuals did not have AML or any disease survival times. The authors should clarify how survival data was used in relation to the AML and non-AML individuals in this combined group.

4. Are there prognostic results from the DREAM9 challenge that this method can be compared to? Or is the main point more about the type of interactions uncovered (which presumably were not uncovered by the DREAM9 challenge participants) moreso than the magnitude of prognostic ability?

5. I was not provided with the R code mentioned in the paper, but I trust that it will be provided with the publication to help people run the method.

**Have all data underlying the figures and results presented in the manuscript been provided?**

Reviewer #1: Yes

Reviewer #2: None

Reviewer #3: Yes

PLOS authors have the option to publish the peer review history of their article (what does this mean?). If published, this will include your full peer review and any attached files.

Reviewer #1: **Yes: **Leo Lahti

Reviewer #2: No

Reviewer #3: No
---

## [Decision Letter · Decision Letter 1]

19 Jan 2021

Dear Dr Sillanpää,

Thank you very much for submitting your manuscript "Model guided trait-specific co-expression network estimation as a new perspective for identifying molecular interactions and pathways" for consideration at PLOS Computational Biology. As with all papers reviewed by the journal, your manuscript was reviewed by members of the editorial board and by several independent reviewers. The reviewers appreciated the attention to an important topic. Based on the reviews, we are likely to accept this manuscript for publication, providing that you modify the manuscript according to the review recommendations.

Reproducibility is important to us. Please make sure your code reproduces all parts of the paper.

Sincerely,

Pejman Mohammadi, Ph.D.

Guest Editor

PLOS Computational Biology

Florian Markowetz

Deputy Editor

PLOS Computational Biology

[LINK]

Reviewer's Responses to Questions

**Comments to the Authors:**

Reviewer #1: Thank you for the improvements. The motivation and technique is now more clearly explained, and reproducible source code provided.

Minor remaining comments:

- I would like to verify that the reproducible source code contains also all code to replicate figures, tables and other such results represented in the manuscript.

- Is an open source license (such as MIT) associated with the code - which license?

If reproducible code for the main results is provided (and not only a demo example of the method itself), and the code is released with an open license for fluent verification and reuse, then I have no further suggestions.

Reviewer #3: The authors have sufficiently addressed my comments.

**Have all data underlying the figures and results presented in the manuscript been provided?**

Reviewer #1: Yes

Reviewer #3: Yes

PLOS authors have the option to publish the peer review history of their article (what does this mean?). If published, this will include your full peer review and any attached files.

Reviewer #1: **Yes: **Leo Lahti

Reviewer #3: No
---

## [Editor Report · Decision Letter 2]

13 Apr 2021

Dear Dr Sillanpää,

We are pleased to inform you that your manuscript 'Model guided trait-specific co-expression network estimation as a new perspective for identifying molecular interactions and pathways' has been provisionally accepted for publication in PLOS Computational Biology.

Best regards,

Pejman Mohammadi, Ph.D.

Guest Editor

PLOS Computational Biology

Florian Markowetz

Deputy Editor

PLOS Computational Biology

---

## [Editor Report · Acceptance letter]

28 Apr 2021

PCOMPBIOL-D-20-01509R2 

Model guided trait-specific co-expression network estimation as a new perspective for identifying molecular interactions and pathways

Dear Dr Sillanpää,

I am pleased to inform you that your manuscript has been formally accepted for publication in PLOS Computational Biology. Your manuscript is now with our production department and you will be notified of the publication date in due course.

With kind regards,

Katalin Szabo
